# Acute influence of an adaptive sporting event on quality of life in veterans with disabilities

**Alexis N. Sidiropoulos**[1]*, **Jonathan J. Glasberg**[1], **Timothy E. Moore**[2], **Leif M. Nelson**[3], **Jason T. Maikos**[1]

1 Department of Veterans Affairs, New York Harbor Healthcare System, New York, New York, United States of America, 2 Statistical Consulting Services, Center for Open Research Resources and Equipment, University of Connecticut, Storrs, CT, United States of America, 3 Department of Veterans Affairs, National Veterans Sports Programs and Special Events, Washington, DC, United States of America

☯ These authors contributed equally to this work.
* alexis.sidiropoulos@va.gov

**Data Availability Statement:** All relevant data and code used to generate plots and perform statistical analyses in this manuscript are available on Zenodo, accessible via: https://zenodo.org/record/7310889.

## Abstract

Veterans with disabilities can experience poor quality of life following military service due to the associated negative physical and psychological ramifications. However, participation in physical activities has shown to induce both physical and mental benefits and improve the quality of life of this population. Adaptive sports, an innovative approach to address the unique physical and psychosocial needs of veterans with disabilities, are becoming more widely used as a rehabilitation tool to improve the quality of life for these veterans. This study aimed to determine the acute influence of participation in a single-day, veteran-based, adaptive kayaking and sailing event on the perceived overall health, quality of life, and quality of social life of veterans with varying disabilities. It was hypothesized that all three categories and the sum score of quality of life would reflect a positive acute response after participation in the community-based physical activity event. Veterans responded to three quality of life-related questions using a 5-point Likert scale before and directly after participating in the event. Findings indicated that an adaptive sporting event can have an acute positive influence on the quality of life of veteran participants, with improvements observed in all three categories of perceived quality of life. Therefore, it is advantageous for the whole-health rehabilitation of veterans with disabilities for the Department of Veterans Affairs to continue to provide opportunities for veterans to participate in non-traditional, community-based activities.

## Introduction

Individuals with disabilities often experience lower quality of life (QoL) compared to those without a disability [1,2]. In particular, veterans with disabilities can experience further diminished QoL due to adverse physical and psychological consequences of military service [3]. However, participation in physical activities for individuals with disabilities has been strongly linked with high QoL, as physical activity provides both physical and mental benefits [4,5]. The physical benefits of participation include advanced cardiac performance, improved

**Funding:** The authors received no specific funding for this work.

**Competing interests:** The authors have declared no competing interests exist.

tendon and muscle strength and flexibility, and higher metabolic capacity [6]. Studies have also indicated significant improvements in psychosocial well-being of individuals with disabilities after participation in physical activities [7], including reduction in post-traumatic stress disorder (PTSD) symptoms [8–10], as well as improved affect and well-being [10].

Veterans with disabilities often face unique challenges. Mental health disorders, often stemming from PTSD, are a primary concern for clinicians at the Department of Veterans Affairs (VA) due to the high prevalence in the veteran population [11]. Tanielian and Jaycox [12] estimated that 31% of veterans returning from Operation Iraqi Freedom (OIF) and Operation Enduring Freedom (OEF) suffer from PTSD. Veterans with physical disabilities, such as amputation, traumatic brain injury (TBI) and spinal cord injury/disorder (SCI/D), are also at risk for adverse psychosocial outcomes, including high rates of initial-onset depression [13], as well as significant levels of anxiety and social isolation [14]. TBI frequently adversely affects emotional well-being, life satisfaction, and community integration [15]. It is estimated that over 200,000 veterans from all eras have been treated across the VA healthcare system with some form of TBI, with over 50,000 of those from OEF and OIF [16]. Individuals with SCI/D can also have compromised aspects of psychosocial well-being, social participation, and overall QoL [17].

Innovative approaches to address the unique physical and psychosocial needs of veterans with disabilities include participation in adaptive sports. Utilizing complementary therapies and modalities to supplement traditional clinical pathways improve rehabilitation outcomes in multiple domains. As such, adaptive sports are a growing rehabilitation tool for individuals with disabilities [18] due to both the physical and psychosocial benefits of participation [19–21] and has been linked to higher QoL [22]. VA has recognized the positive correlation between QoL and participation in recreational activities and has made adaptive sports a major focus of non-traditional clinical care in recent years [18].

Compared to sedentary individuals with disabilities, those who participate in adaptive sports indicate better QoL, greater happiness and general satisfaction with their life, improved vitality and cardiovascular health, and a substantially increased self-efficacy [23–25]. Large-scale, week-long rehabilitation events held by VA, such as the National Veterans Wheelchair Games, which is a multi-event sporting program for veterans who use wheelchairs, and the National Disabled Veterans Winter Sports Clinic, an adaptive skiing event for veterans with disabilities, report increased physical health and perceived QoL from participants [26]. Individuals who participate in adaptive sports experience fewer days of pain, anxiety, depression, and sleeplessness, and less severe secondary health conditions [2], many of which are symptoms commonly experienced by veterans [19]. However, a gap in the literature remains, as the acute response to such events has yet to be evaluated. This potential response is important to identify, as it can allow veterans and other individuals with disabilities to experience an improvement in QoL without the required commitment of long-term physical activity. Commitment to participate in a one-day adaptive sporting event may provide an opportunity for more veterans to experience an improvement in QoL, as a one-day event may be more feasible in terms of logistics and motivation for these individuals.

The aim of this study was to determine the acute influence of participation in a single-day, veteran-based, adaptive kayaking and sailing event on QoL in three categories: Perceived Overall Health, Perceived QoL, and Perceived Quality of Social Life (QoSL). Veterans were asked to respond to qualitative surveys regarding their perceived QoL before and after participation in a group adaptive sporting event. It was hypothesized that a positive acute response would be evidenced in all three categories and a QoL sum score after veterans with disabilities participated in a community-based physical activity event.

## Methods

### Participants

This study was a retrospective medical-chart review and included data from veterans who participated in a one-day adaptive kayaking and sailing event, Heroes on the Hudson, in either 2018 or 2019. This retrospective analysis was approved by the VA New York Harbor Healthcare System Institutional Review Board (IRB). A waiver of documentation of informed consent was granted by the IRB due to the retrospective nature of the data analysis. Therefore, no written or verbal consent was obtained from veteran participants. Veterans of all abilities were eligible to participate and all veterans who registered for the event were enrolled. Medical diagnoses included: amputation of the upper or lower limb, visual impairment, SCI/D, TBI, mental health conditions, and general, non-classified medical conditions. All participants received medical clearance from their primary care or rehabilitation specialty care provider prior to participation.

### Adaptive kayaking and sailing event

Heroes on the Hudson is an annual one-day adaptive maritime sporting event for New York and New Jersey veterans with disabilities designed to progress rehabilitation through sports therapy and therapeutic recreation out of the clinic and into the community. This study includes data collected from two events (2018 and 2019). During that time, individuals with disabilities received interdisciplinary care, which included professional instruction on sailing and kayaking through VA community partners, along with support from associated VA clinical and medical providers. Participants socialized and engaged in two hours of kayaking and sailing each. Sailing activities consisted of groups of four or five participants per sailboat, while participants used single or tandem kayaks as part of a larger group for kayaking activities. All equipment and instruction were modifiable to support the veterans of varying abilities.

### Outcome measures

Demographic and injury-related variables were collected, including age, sex, military service era, and disability diagnosis. The QoL survey implemented in this study represents a modified subsection of a validated survey (The Influence on Quality of Life Scale) that evaluates the influence of QoL in athletes with disabilities [2,27]. The instrument (Fig 1) asked veterans to

| Over the past month, how would you rate your: | Very Poor | Poor | Neither Poor Nor Good | Good | Very Good |
|---|---|---|---|---|---|
| Overall Health | | | | | |
| Quality of Life | | | | | |
| Quality of Social Life | | | | | |

| After participating in Heroes on the Hudson, how would you rate your: | Very Poor | Poor | Neither Poor Nor Good | Good | Very Good |
|---|---|---|---|---|---|
| Overall Health | | | | | |
| Quality of Life | | | | | |
| Quality of Social Life | | | | | |

**Fig 1. Survey utilized pre- and post-participation in Heroes on the Hudson event.**

rate, on a 5-point Likert scale, the influence their participation had on 3 items: perceived overall health, perceived QoL, and perceived QoSL. Responses could range from "Very Poor" to "Very Good". Validity and reliability were calculated for a combined QoL sum score comprised of the three items. Additionally, previous literature has shown that use of this survey for athletes with disabilities showed evidence of construct validity and an acceptable internal consistency (alpha = 0.87) [2]. Hard copies of surveys were distributed at the event site to participants both pre- and post-participation in the Heroes on the Hudson event.

## Statistical analysis

To evaluate the impact of the event on participant QoL, a combined QoL sum score of the three individual items was calculated for the pre- and post-surveys of each participant. A one-factor Confirmatory Factor Analysis (CFA) was used to test validity and reliability of the QoL sum score. The QoL sum score was then analyzed for differences in pre-post survey scores using Wilcoxon signed rank tests. Additionally, pre-post scores for individual items were also compared using Wilcoxon signed rank tests. Furthermore, an Analysis of Covariance (ANCOVA) model was used to evaluate which participant characteristics correlated with QoL. This model predicted the post-survey QoL sum score of participants using the following predictors: Diagnosis, Military Service Era, Age, Sex, and Pre-survey QoL sum score. The increased flexibility in the relationship between the pre- and post-survey QoL sum scores allows this type of regression model to be more effective than other possible methods (e.g., using change in score as a response) [28]. ANCOVA models were also fitted separately on each QoL item. The proportion of variance explained by each predictor ($\eta^2$, a measure of the amount of variance in a dependent variable that can be explained by one or more independent variables) was used as a measure of the effect size for each of the model predictors. All statistical tests were performed in R 4.1.0 (R Core Team, 2021) [29] and significance was set at $p < 0.01$.

## Results

### Validity and reliability

The QoL sum score indicated good reliability ($\alpha_{rel}$ = 0.9) and validity (average variance extracted, $AVE$ = 0.85). All three items loaded significantly onto the single latent variable (Standardized Coefficients > 0.85, $p$ <0.0001). The average inter-item correlation, based on Spearman Rank correlations for the pre-survey QoL sum score, was 0.77, and 0.6 for the post-survey QoL sum score, indicating a relatively high overlap between questions. Overall, these results suggest that combining the single QoL items into a summed QoL score is appropriate.

### Participant characteristics

Between 2018–2019, 92 veterans participated in a Heroes on the Hudson event and 58 of those participants completed both the pre- and post-survey, indicating that 63% of participants were willing and able to answer the survey questions. Table 1 describes the participant characteristics for those that provided all requested demographic information (n = 52). ANCOVA results suggested no differences in participant characteristics were identified between QoL sum scores ($p$ > 0.01 for all demographic covariates). However, when individual questions were evaluated using ANCOVA, a significant difference between males and females ($F_{1,36}$ = 6.37, $p$ = 0.02) indicated that males, on average, had higher post-survey Overall Health scores than females. However, sex only explained relatively little (5%, $\eta^2$ = 0.05) of the variation in post-survey score. No significant differences in demographic variables were identified for QoL or QoSL.

**Table 1. Participant characteristics.**

| Characteristics | Number of Participants |
|---|---|
| **Military Service Era** | |
| Vietnam/Korea | 18 (34.6%) |
| OEF[a]/OIF[b] | 20 (38.5%) |
| Post-Vietnam | 6 (11.5%) |
| Gulf | 8 (15.4%) |
| **Diagnosis** | |
| Mental Health | 20 (38.5%) |
| Amputation | 5 (9.6%) |
| Visual Impairment | 5 (9.6%) |
| General Medical | 9 (17.3%) |
| SCI/D[c] | 7 (13.5%) |
| TBI[d] | 5 (9.6%) |
| Missing | 1 (1.9%) |
| **Sex** | |
| Male | 42 (80.5%) |
| Female | 10 (19.2%) |
| **Age** | |
| Mean(Standard Deviation) | 57.0 (15.0) |
| Median [Min, Max] | 57.0 [27.0, 87.0] |
| Missing | 3 (5.8%) |

[a]OEF: Operation Enduring Freedom.

[b]OIF: Operation Iraqi Freedom.

[c]SCI/D: Spinal Cord Injury/Disorder.

[d]TBI: Traumatic Brain Injury.

## Pre-post survey score analysis

QoL sum scores increased from pre- to post-event, as indicated by an increase in the median and mean values in the group after participation (Fig 2), which was supported with a Wilcoxon signed rank test (Z = -4.09, $p$ <0.0001). Individual items of Overall Health (Z = -3.27, $p$ = 0.0011), QoL (Z = -3.18, $p$ = 0.0015), and QoSL (Z = -3.61, $p$ = 0.0003) also indicated significant improvements after participating in the adaptive sporting event (Fig 3).

Table 2 describes the acute changes in influence of each individual item: Overall Health (Table 2A), QoL (Table 2B) and QoSL (Table 2C). The diagonal dark gray area highlights the individual scores that did not change from pre- to post-survey (e.g., pre-score of 3 and a post-score of 3). Scores above the diagonal, shown in light gray, represent those that improved post-event (e.g., pre-score of 2 and a post-score of 4). Scores below the diagonal, without highlight, represent those that decreased post-event (e.g., pre-score of 4 and post-score of 3). Note that most individuals either did not change or improved their scores after participating in the event. For example, eight participants reported an increase from four to five in the QoL item.

## Sum score analysis

The ANCOVA model fit the data well, displaying normally distributed residuals and explaining a statistically significant and substantial portion of the variance in post-event sum score ($F_{11,36}$ = 5.84, $p$ <0.0001, $R^2$ = 0.64, $R^2_{adj}$ = 0.53). Pre-survey QoL sum score was the only significant predictor of post-survey QoL sum score ($p$ <0.0001), and it explained 29% ($\eta^2$ = 0.29)

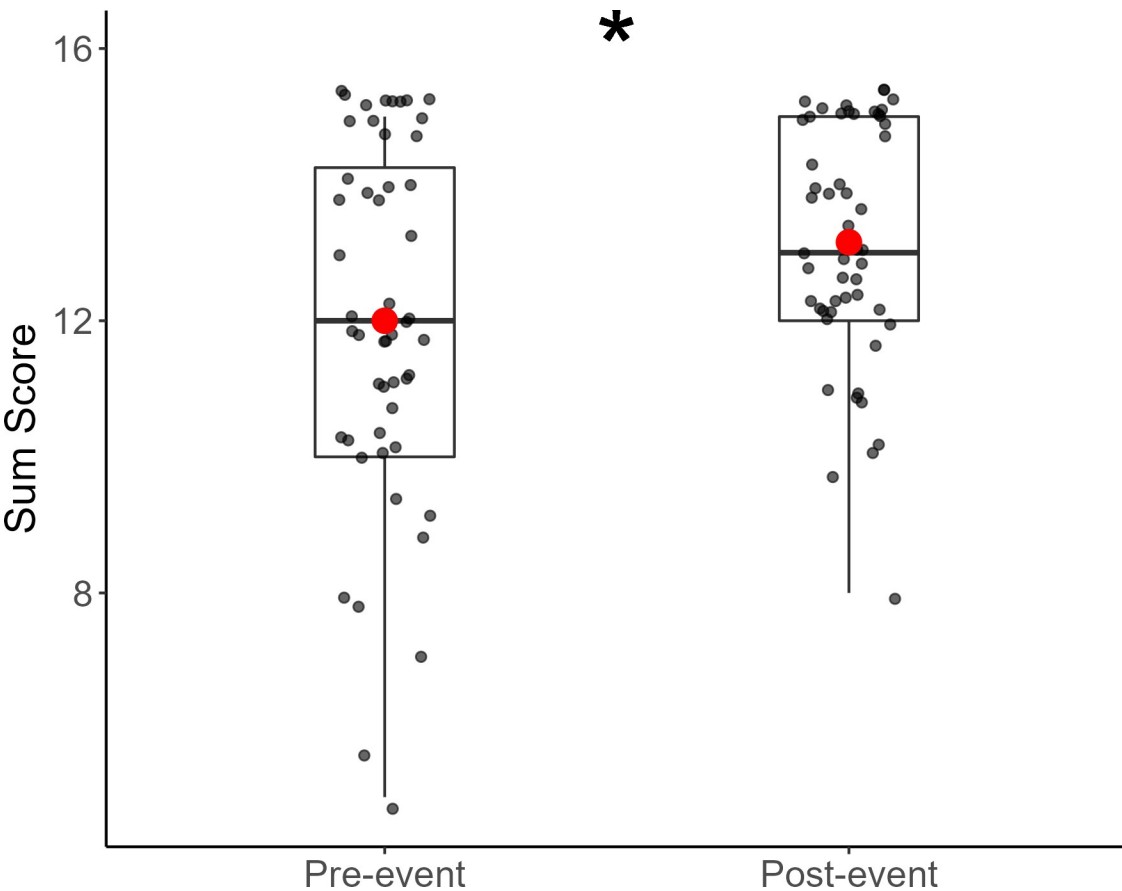

**Fig 2. Boxplot of Pre- and Post-QOL sum scores.** Points show individual participant data, jittered for clarity. Red dots represent means. Asterisks represent significant differences between pre- and post-survey results.

of the variance in post-survey score. The only other variable that contributed substantially to the variance explained in post-participation QoL sum score was Diagnosis, which explained an additional 11%, but was not statistically significant ($p = 0.08$).

### Individual item analysis

Similar to the QoL sum scores, the strongest predictors of the individual item post-survey responses were pre-survey scores, demonstrated by the proportion of variance they explained (Overall Health: 35%, QoL: 24%, QoSL: 15%). For individual items, the only significant demographic predictor was Sex, which was a predictor for Overall Health post score. A marginally significant difference between males and females ($F_{1,36} = 6.37$, p = 0.02) was identified, with males having on average, higher post-participation Overall Health scores when adjusting for pre-participation score, age, diagnosis and military service era. Sex, however, only explains 5% of the variation in post-participation.

### Discussion

Veterans experienced an acute positive influence on QoL after participation in an adaptive sporting event. In support of our hypothesis, improvements in perceived overall health, QoL, QoSL, and the QoL sum score were observed after participating in Heroes on the Hudson. Additionally, this study represents a first attempt at identifying key participant characteristics

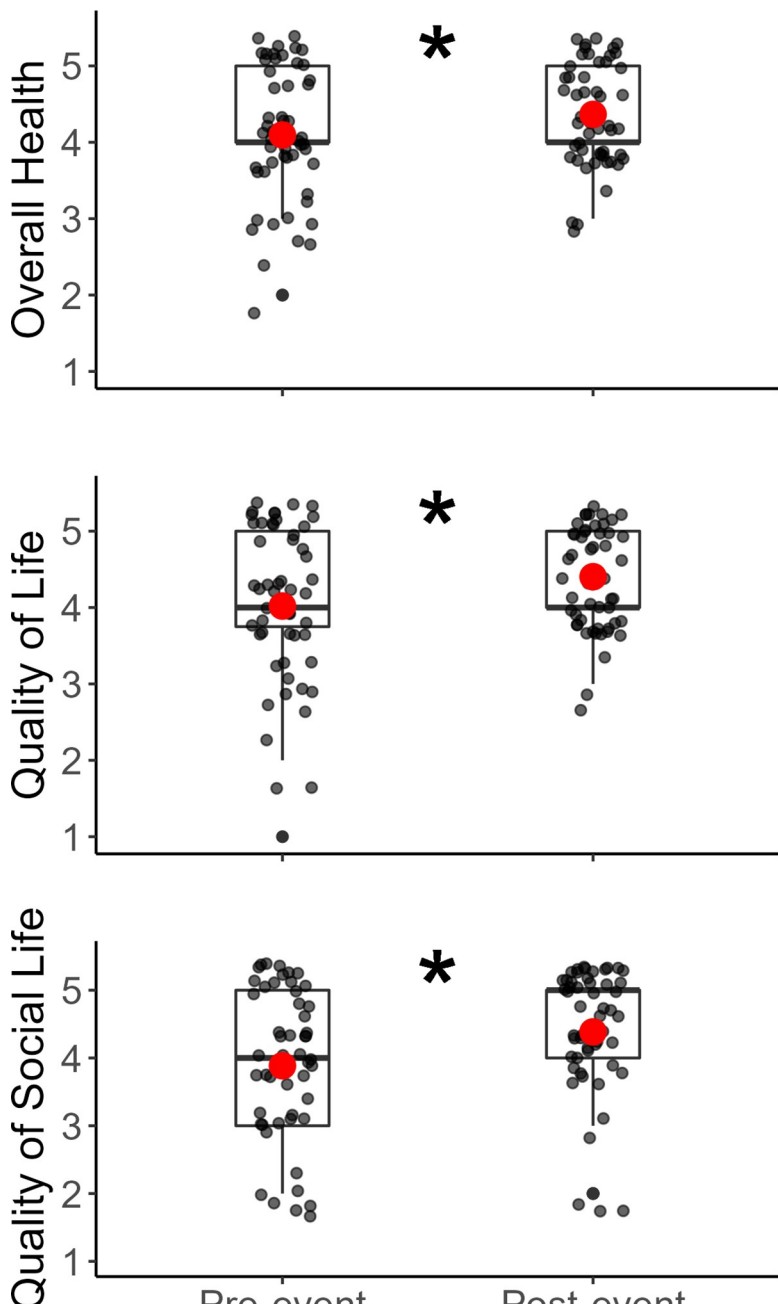

**Fig 3. Pre- and post-scores on the acute influence of participation in the Heroes on the Hudson event on Overall Health, QoL, and QoSL.** QoL: Quality of Life. QoSL: Quality of Social Life. Asterisks represent significant differences between pre- and post-survey results. Points show individual participant data, jittered for clarity. Red dots represent means. Asterisks represent significant differences between pre- and post-survey results.

that significantly correlate with QoL. There were no significant (at $\alpha = 0.01$) associations between participant characteristics (Sex, Diagnosis, Age, Gender, and Military Service Era) and QoL outcomes, for either the QoL sum score or individual items. One possible explanation for this result may be due to the need to aggregate data for the smaller subgroups, particularly for diagnosis and military service era, which limited the ability to explore interaction terms in the regression model. Future analyses, with larger sub-category sample sizes, will be critical to

**Table 2. Survey responses.**

| A. Overall Health Score | | Post-Survey Scores | | | | |
|---|---|---|---|---|---|---|
| | | 1 | 2 | 3 | 4 | 5 |
| Pre-Survey Scores | 1 | 0 | 0 | 0 | 0 | 0 |
| | 2 | 0 | 0 | 1 | 1 | 0 |
| | 3 | 0 | 0 | 3 | 6 | 0 |
| | 4 | 0 | 0 | 0 | 17 | 6 |
| | 5 | 0 | 0 | 0 | 1 | 17 |
| **B. Quality of Life Score** | | **Post-Survey Scores** | | | | |
| | | 1 | 2 | 3 | 4 | 5 |
| Pre-Survey Scores | 1 | 0 | 0 | 0 | 1 | 0 |
| | 2 | 0 | 0 | 0 | 3 | 0 |
| | 3 | 0 | 0 | 2 | 7 | 0 |
| | 4 | 0 | 0 | 1 | 11 | 8 |
| | 5 | 0 | 0 | 0 | 3 | 16 |
| **C. Quality of Social Life Score** | | **Post-Survey Scores** | | | | |
| | | 1 | 2 | 3 | 4 | 5 |
| Pre-Survey Scores | 1 | 0 | 0 | 0 | 0 | 0 |
| | 2 | 0 | 3 | 0 | 2 | 2 |
| | 3 | 0 | 0 | 1 | 5 | 3 |
| | 4 | 0 | 0 | 1 | 11 | 7 |
| | 5 | 0 | 0 | 0 | 1 | 16 |

help identify any true effects and make comparisons between participants with different characteristics. This will also allow for a greater refined grouping of participants (e.g., PTSD versus other mental health conditions), as well as the inclusion of potential interaction effects (e.g., sex and diagnosis).

Despite the small subgroup sample sizes, these results also indicated that no single sub-category was a driver for improvements in QoL, suggesting that, on average, all groups benefited equally from participation in this event regardless of diagnosis, military service era, or other unique characteristic. However, by identifying key characteristics that are predictive of changes in QoL for adaptive sporting events, future events and activities can be designed or tailored to ensure all groups of veterans, regardless of unique characteristics, can benefit to their greatest extent from participation. As such, more research will be critical in identifying which characteristics are most associated with QoL outcomes. Inclusion of follow-up questions to determine any causes of positive or negative change in QoL can also help inform future events.

## Participation in an adaptive sporting event positively influences QoL

It is well-established that participation in sport activity can positively influence an individual's perception of QoL [2]. For individuals with disabilities, physical activity can lead to feelings of functional efficiency and accomplishment, which can then foster empowerment as a response to the challenge of the activity [30]. These individuals may also develop personally empowering skills that can enhance effectiveness in other life situations [30]. Specifically, engagement in sport activities may enable individuals with disabilities to improve efficacy in activities of daily living, which can lead to further feelings of self-efficiency and improvements in QoL [23].

Social aspects of QoL for those with disabilities are also enhanced during group participation in physical activities with other participants with disabilities, as a feeling of interpersonal inclusiveness is promoted instead of isolation, which is commonly experienced by this

population [30]. Due to the social value of physical activity, participation in such activities provides these individuals an opportunity to feel more integrated into society, as previous literature has reported that participation has granted them "social acceptability" [30]. This finding supports the increase in QoSL identified in this study, indicating that veterans experienced a social benefit from their participation in the Heroes on the Hudson event.

## Veteran-specific advantages of participation in a physical activity event

While participating in adaptive sport activities can be difficult for individuals with disabilities, previous research has reported that limited resources are one of the major barriers of physical activity for these individuals, specifically veterans with disabilities [31,32]. Therefore, removal of barriers for adaptive sports participation is imperative to the goal of improving the QoL of sedentary individuals. When asked, veterans noted that offering physical activities with exercise partners or group classes hosted by VA would positively influence their decision to participate in such activities [10]. This increased opportunity to access physical activities can have a major perceived impact on veterans' personal lifestyles [10], highlighting the importance of removing such barriers.

Specifically, elimination of such constraints for veterans with disabilities requires greater attention, since social isolation, commonly observed in this population, can be partly due to PTSD-related symptoms [10]. Participation in an organized sporting event can decrease anxiety and provide motivation to participate in other sporting activities [26]. Reported confidence gained by participation in these types of events have been partially attributed to the camaraderie veterans maintain with each other, as it has been shown that individuals who have commonalities are typically more comfortable with one another [26,31]. Previous research has also indicated that the increased social interaction that occurs during a physically active event with other veterans is viewed positively by veterans and can stimulate social meetings outside of the event [10]. Therefore, these experiences also provide opportunities for advancing social skills and broadening social experiences for individuals with disabilities [30]. Events such as Heroes on the Hudson, may act as an incentive for veterans with disabilities to partake in physical activities that they would not have participated in otherwise. Consequently, involvement in Heroes on the Hudson, which provides veterans with exercise partners and is hosted by the VA, can positively influence their acute QoL, as evidenced in this study.

Participation in an event such as Heroes on the Hudson also highlights the advantages of community-based activities specifically designed for veterans with disabilities. Rehabilitation for individuals with disabilities has long focused on clinic-based physical improvements; however scientific literature has recognized that both social and community reintegration are important components of rehabilitation for this population [33]. This reintegration includes a sense of belonging or acceptance, connection to other people, and involvement in leisure and community activities [34]. Community reintegration, which is especially challenging for injured veterans, is one of the ultimate goals of rehabilitative efforts [35]. The Military to Civilian Questionnaire (M2C-Q), which characterizes veterans' perceived difficulties reintegrating back into the community life following combat [35] and the Community Reintegration for Service Members (CRIS) survey [36], which assesses community integration issues from a service person's perspective, may prove to be imperative tools in this population [34]. These surveys may also provide vital information on how best to implement physical community activities to improve community integration in disabled veterans if administered in conjunction with an event. Notably, social participation is also a focus of the Heroes on the Hudson event and an important factor in the prediction of QoL among individuals with disabilities [32]. Both community reintegration and social participation are inherent features of the

Heroes on the Hudson event, and therefore may have elevated the positive acute improvement observed in all three survey categories included in this study.

More specifically, the therapeutic potential of adaptive sports and recreation can address the needs of returning combat veterans who have acquired a disability [19], as such activities have been shown to increase perceived competence and a general sense of vigor. Previous research highlighted that physical activity positively influences mental well-being of individuals by improving emotion and mood, self-esteem, and cognitive dysfunction, while also decreasing symptoms of depression and anxiety [37]. This information is directly applicable to the population of veterans in this study, as the Mental Health subgroup was the largest of all medical conditions. Importantly, the causal link between participation in physical exercise and decreased depression has indicated that the antidepressant effect of exercise can be of a similar magnitude to that of other psychotherapy interventions [38], therefore potentially reducing the need for other, more invasive interventions. This can be useful in veterans with PTSD, who often exhibit reduced participation in regular physical activity compared to pre-PTSD time periods [39]. It has been suggested that this lack of activity contributes to the medical, cognitive, and neural comorbidities associated with sedentary behavior [39]. Previous literature has indicated that exercise may reduce PTSD symptomology in veterans, as it is associated with multiple psychological, cognitive, and neurobiological benefits and lacks the stigma of standard mental health treatment [39,40].

## Long-term effects of participation

While the acute positive influence on QoL observed in this study promotes participation in events such as Heroes on Hudson, extended periods of participation in exercise and adaptive sporting activities may be associated with longer effects on QoL in veterans with disabilities. Previous literature supports this notion, as a similarly designed study investigating the impact of a 10-day adaptive sporting event on physical and psychological outcomes in veterans with disabilities indicated a rebound effect three months post-event [41]. However, longer duration of activity participation has shown lasting effects, as a study on individuals with fibromyalgia who participated in a 10-week group exercise program reported improved QoL at the 6-month follow-up [42]. Additionally, mobility-challenged older adults who participated in a 6-month exercise program reported improvements in QoL at a 5-year follow-up [43] and patients with chronic heart failure who participated in a year-long exercise program reported improved QoL in a 3-year follow up assessment [44]. While acute improvement in QoL is noteworthy in veterans with disabilities, lasting effects on perceived QoL may not be detected following a one-day adaptive sporting event. Results of this study, in combination with previous literature, support the expansion of these non-traditional community-based activities to supplement traditional clinical pathways and improve whole-health rehabilitation of veterans with disabilities. Fortunately, VA provides grant opportunities to qualifying community-based organizations to bolster programs that provide veterans with longer-term adaptive sports opportunities, allowing veterans to continue their participation in adaptive sports to improve and maintain their new level of independence, well-being, and QoL. Future research should aim to determine if the acute responses illustrated in this study can also be observed as long-term effects of participation in similar adaptive sporting events.

While physical exercise can reduce long-term anxiety tendencies, a single session has been shown to decrease short-term anxiety, anger, and tension [19]. Therefore, participation in single-session events similar to Heroes on the Hudson can positively improve mood, as evidenced by the findings of this study. Given a larger sample size, the results of this study may statistically support this notion. Consequently, an invitation and encouragement to participate in

these types of events may be imperative to veterans returning from military service with disabilities for both their physical and mental rehabilitative endeavors.

## Limitations and future research

This study focused on the acute impact of a community-based event on QoL using non-specific questions. Future research would benefit from including follow-up surveys (i.e., three- or six-months post-event) to determine the long-term effects of participation in a one-day adaptive sporting event. Further, more detailed information can be obtained with the use of more targeted questions regarding the emotional and mental outcomes of participation in this type of event, such as the Medical Outcomes Study Short Form 36-Item Health Survey. This survey is a measure of health-related QoL, as it spans physical and mental health, is reliable and valid in ambulatory individuals, and utilized in veteran patients [45]. Additionally, a larger sample size to accommodate the many sub-groups of veteran status and diagnosis would improve the utility of the study results and allow more precise estimations of the effects of participant characteristics on QoL outcomes. Although the percentage of women participants in this study (23%) represents double the current percentage of female veterans receiving care at VA (10.5%) [46], future adaptive sporting events should aim to invite and include more women participants to heighten the generalizability of the results. Lastly, this event was available only to those in more suburban communities in New York and New Jersey; therefore, results may not be generalizable to more rural parts of the country.

Future research on this topic should also aim to determine if veterans continued to participate in other adaptive sporting events within VA, with VA community partners, or other organizations within their community. This information can help identify the requirements for a long-term positive influence on the QoL of this population. Further, previous literature has noted that inclusion of family members in physical activity events also helps to improve quality of family life [27]. Thus, by inviting family members to participate alongside the veterans, the positive outcomes created by the event may extend beyond the direct influence on the individuals and expand into the family dynamic as well.

## Conclusions

Participation in a community-based adaptive sports event, such as Heroes on the Hudson, led to an acute, positive influence on the QoL of individuals with disabilities. Previous research has reported that the act of participation in such an event impacts empowerment of these individuals regardless of the activity type [30]; however, due to the limited opportunities to experience outcomes associated with a physical activity event alongside other disabled individuals, the impact of participation holds even greater importance for the wellbeing of this population [32]. Though lack of statistical power did not allow consideration of interaction effects, the absence of statistical significance should not discourage these research efforts. Instead, this should be used as a reason to collect more data to help in identifying participant characteristics that are most strongly associated with observed changes in QoL. Further, a lack of statistical significance informs VA that the adaptive sporting events are helping to improve the QoL in all veterans regardless of military service era, diagnosis, or sex. Thus, it is advantageous for the whole-health rehabilitation of veterans with disabilities to participate in non-traditional, community-based activities.

## Acknowledgments

The authors would like to thank the veterans for their participation in the event and survey. Additionally, the authors thank the volunteers and community partners, including Hudson

River Community Sailing and Inwood Canoe Club, for their help in creating Heroes on the Hudson.

## Author Contributions

**Conceptualization:** Jonathan J. Glasberg, Leif M. Nelson, Jason T. Maikos.

**Data curation:** Alexis N. Sidiropoulos, Jonathan J. Glasberg, Leif M. Nelson, Jason T. Maikos.

**Formal analysis:** Alexis N. Sidiropoulos, Timothy E. Moore.

**Software:** Alexis N. Sidiropoulos.

**Supervision:** Jason T. Maikos.

**Validation:** Timothy E. Moore.

**Visualization:** Timothy E. Moore.

**Writing – original draft:** Alexis N. Sidiropoulos.

**Writing – review & editing:** Jonathan J. Glasberg, Leif M. Nelson, Jason T. Maikos.

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
