## [Decision Letter · Decision Letter 0]

24 Jun 2022

PONE-D-22-10455Acute influence of an adaptive sporting event on quality of life in veterans with disabilitiesPLOS ONE

Dear Dr. Sidiropoulos,

Thank you for submitting your manuscript to PLOS ONE. After careful consideration, we feel that it has merit but does not fully meet PLOS ONE’s publication criteria as it currently stands. Therefore, we invite you to submit a revised version of the manuscript that addresses the points raised during the review process.

We look forward to receiving your revised manuscript.

Kind regards,

Yih-Kuen Jan, PhD

Academic Editor

PLOS ONE

Journal Requirements:

Reviewers' comments:

Reviewer's Responses to Questions

**Comments to the Author**

1. Is the manuscript technically sound, and do the data support the conclusions?

Reviewer #1: No

Reviewer #2: Yes

2. Has the statistical analysis been performed appropriately and rigorously? 

Reviewer #1: No

Reviewer #2: Yes

3. Have the authors made all data underlying the findings in their manuscript fully available?

Reviewer #1: No

Reviewer #2: Yes

4. Is the manuscript presented in an intelligible fashion and written in standard English?

Reviewer #1: Yes

Reviewer #2: Yes

5. Review Comments to the Author

Reviewer #1: Title “Acute influence of an adaptive sporting event on quality of life in veterans with disabilities “is not a interesting topic for veterans with disabilities. However, there are the following concerns in this manuscript.

1、Only 58 of those participants completed both the pre-and post-survey, 27% participants refused to answer the survey, please explain the reasons.

2、Veterans who participated in only a one-day adaptive kayaking and sailing event, time is too short. How to figure out the other reasons may lead to the differences about quality of life. Please explain it.

3、The Influence on Quality of Life Scale construct validity should be reported. Quality of life for athletes with cerebral palsy may do not suit for veterans with disabilities.

4、In the analysis, the manuscript should add age, sex and other variable as confounder variable in the analysis.

Reviewer #2: The authors conducted a retrospective analysis of questionnaire data, proving that the perceived QoL of disabled veterans is immediately improved after these veterans participate in a single-day Veteran-based adaptive kayaking and sailing event. This study highlights the value of encouraging disabled veterans to participate in adaptive sports and is suitable for publication, only if the authors respond to the following questions:

1. In Line54–63 (INTRODUCTION), it seems better to pinpoint the "gap", the research gap which previous studies have left but your research will fill. For example, your research highlights the ACUTE influence of participation in A SINGLE-DAY, so is this an aspect that previous studies ignore but your research notice? If so, please elaborate on this "gap", which seems one of your novelty worth highlighting.

2. In DISCUSSION, to corroborate the current study, the authors EXTENSIVELY discuss the previous research (esp. on whether adaptive sports improve the mental health & QoL of the disabled), but it seems better to add extra information about the current study's RESULTS, among which several interesting details seem worth further discussing/explaining:

2.1 As the authors mentioned in Line148 & Table2, after participation, the OEF/OIF veterans had the greatest positive change scores, compared with veterans during other military service eras. This data is so striking that it seems better to briefly explain the possible reasons and implications for this trend.

2.2 In Table2, after participation, a decrease of QoL/QoDL was reported by veterans with orthopedic or spinal cord injury. This data is eye-catching and worth a brief discussion/explanation.

3. In REFERENCES, over 91% (43/47) of references have been published for over 5 years, thus seeming "old". It seems better to cite more RECENT references (published after 2016). Since 2016, there have been at least 60 papers in the field of "("adaptive sport*") AND disab*", such as PMID: 31046012, PMID: 32427531, and PMID: 32427531.

6. PLOS authors have the option to publish the peer review history of their article (what does this mean?). If published, this will include your full peer review and any attached files.

Reviewer #1: No

Reviewer #2: **Yes: **Zewei Zhang

---

## [Author Response · Author response to Decision Letter 0]

3 Oct 2022

September 29, 2022

Dr. Yih-Kuen Jan

Academic Editor

PLOS ONE

Subject: Revision and resubmission of manuscript PONE-D-22-10455

Dear Dr. Yih-Kuen Jan,

Thank you for your email and the opportunity to revise our manuscript entitled “Acute Influence of an Adaptive Sporting Event on Quality of Life in Veterans with Disabilities”. We greatly appreciate the reviewers for their insightful comments. The manuscript has been improved after making the suggested edits. 

Please find attached a point-by-point response to the reviewer’s concerns. The revision has been developed in consultation with all coauthors. We hope that you find our responses satisfactory, and that the manuscript is now acceptable for publication. We thank you for your continued interest in our research. 

Sincerely,

Alexis Sidiropoulos, PhD

Department of Veterans Affairs

New York Harbor Healthcare System

 

Journal Requirements Addressed:

Comment 1: PLOS ONE style requirements for file naming. 

Response: Following the guidance, we have created separate files for manuscript figures with appropriate file types and names.

Comment 2: Statement of where the minimal data set underlying the results described in the manuscript can be found. 

Response: We have included information in the Data Availability Statement indicating that the minimal data set underlying the results described in this manuscript can be found in Dryad Digital Repository. Dryad is a curated resource that makes the data underlying scientific publications discoverable, freely usable, and citable while promoting openly available research integrated with scholarly literature.

Comment 3: Please include your full ethics statement in the ‘Methods’ section of your manuscript file. In your statement, please include the full name of the IRB or ethics committee who approved or waived your study, as well as whether or not you obtained informed written or verbal consent. If consent was waived for your study, please include this information in your statement as well. 

Response: We have included additional statements within the ‘Methods’ section to clearly state that the Department of Veteran Affairs New York Harbor Healthcare System Institutional Review Board waived the consent of participants for this study due to its retrospective nature. 

Reviewer 1:

Comment 1: Only 58 of those participants completed both the pre- and post-survey, 27% participants refused to answer the survey, please explain the reason. 

Response: We thank the reviewer for their note about the percentage of event participants that did not complete the survey. We would like to note that none of the participants refused, however there were many reasons why these participants did not complete the survey. Some examples include beginning the event without having completed the pre-survey and leaving the event without knowing that post-survey information was requested. 

Comment 2: Veterans who participated in only a one-day kayaking and sailing event, time is too short. How to figure out the other reasons may lead to the differences about quality of life. 

Response: In agreement with the reviewer, we list this short time period as a limitation in our study. Further, we suggest that future research should include a follow-up survey to determine if the positive acute effects on quality of life are retained long-term. Please see the “Limitation and future research” section. 

Comment 3: The Influence on Quality of Life Scale construct validity should be reported. Quality of life for athletes with cerebral palsy may do not suit for veterans with disabilities.

Response: In agreement with the reviewer, statistical analyses were conducted to confirm the validity and reliability of the survey results. This information is included an additional sub-section within the “Results” termed “Validity and Reliability”. 

Comment 4: In the analysis, the manuscript should add age, sex and other variable as confounder variable in the analysis. 

Response: Additional statistical analyses have been conducted to identify any confounding participant characteristics. Further, a regression model fit to the data determined the contribution of each participant characteristic to the survey sum score. The additional statistical analyses are reported in the second paragraph of the “Statistical Analysis” sub-section of the “Methods” section. Please see the sub-sections under the “Results” sections entitled “Participant Characteristics” and “Sum Score Analysis”. 

Reviewer 2:

Comment 1: In Line 54-63 (INTRODUCTION), it seems better to pinpoint the “gap”, the research gap which previous studies have left but your research will fill. For example, your research highlights the ACUTE influence of participation in A SINGLE-DAY, so is this an aspect that previous studies ignore but your research notice? If so, please elaborate on this “gap”, which seems one of your novelty worth highlighting.

Response: We thank the reviewer for this comment and have added information into the introduction to address the gap in the literature which this research fills. Please see lines 59-65.

Comment 2: In DISCUSSION, to corroborate the current study, the authors EXTENSIVELY discuss the previous research (esp. on whether the adaptive sports improve the mental health & QoL of the disabled), but it seems better to add extra information about the current study’s RESULTS, among which several interesting details seem worth further discussing/explaining:

2.1: As the authors mentioned in Line 148 & Table 2, after participation, the OEF/OIF veterans had the greatest positive change scores, compared with veterans during other military service eras. This data is so striking that it seems better to briefly explain the possible reasons and implications for this trend. 

2.2: In Table 2, after participation, a decrease of QoL/QoSL was reported by veterans with orthopedic or spinal cord injury. This data is eye-catching and worth a brief discussion/explanation.

Response: Following the updated statistical analyses, previously discussed trends were no longer valid. However, we have included an updated discussion section from lines 196-217 to discuss the adjusted findings. 

Comment 3: In REFERENCES, over 91% (43/47) of references have been published for over 5 years, thus seeming “old”. It seems better to cite more RECENT references (published after 2016). Since 2016, there have been at least 60 papers in the field of “(adaptive sport*”) AND disab*, such as PMID: 31046012, PMID: 32427531, and PMID: 3247531.

Response: In agreement with the reviewer, we have updated 11 citations in the “References” section with more recent publications. However, due to the limited research published in Veteran- and amputee-specific adaptive sports programs, some “older” references remain in the manuscript. We believe this highlights the need for further research in this field.

---

## [Decision Letter · Decision Letter 1]

6 Nov 2022

Acute influence of an adaptive sporting event on quality of life in veterans with disabilities

PONE-D-22-10455R1

Dear Dr. Sidiropoulos,

We’re pleased to inform you that your manuscript has been judged scientifically suitable for publication and will be formally accepted for publication once it meets all outstanding technical requirements.

Kind regards,

Yih-Kuen Jan, PhD

Academic Editor

PLOS ONE

Additional Editor Comments (optional):

Reviewers' comments:

Reviewer's Responses to Questions

**Comments to the Author**

1. If the authors have adequately addressed your comments raised in a previous round of review and you feel that this manuscript is now acceptable for publication, you may indicate that here to bypass the “Comments to the Author” section, enter your conflict of interest statement in the “Confidential to Editor” section, and submit your "Accept" recommendation.

Reviewer #1: All comments have been addressed

Reviewer #2: All comments have been addressed

2. Is the manuscript technically sound, and do the data support the conclusions?

Reviewer #1: Yes

Reviewer #2: Yes

3. Has the statistical analysis been performed appropriately and rigorously? 

Reviewer #1: Yes

Reviewer #2: Yes

4. Have the authors made all data underlying the findings in their manuscript fully available?

Reviewer #1: Yes

Reviewer #2: Yes

5. Is the manuscript presented in an intelligible fashion and written in standard English?

Reviewer #1: Yes

Reviewer #2: Yes

6. Review Comments to the Author

Reviewer #1: The revised manuscript have improved a lot. All authors did a lot of hard wort to revised it. I just hope to recruite more subjects in the future study.

Reviewer #2: (No Response)

7. PLOS authors have the option to publish the peer review history of their article (what does this mean?). If published, this will include your full peer review and any attached files.

Reviewer #1: No

Reviewer #2: **Yes: **Zewei Zhang

---

## [Editor Report · Acceptance letter]

17 Nov 2022

PONE-D-22-10455R1 

Acute influence of an adaptive sporting event on quality of life in veterans with disabilities 

Dear Dr. Sidiropoulos:

I'm pleased to inform you that your manuscript has been deemed suitable for publication in PLOS ONE. Congratulations! Your manuscript is now with our production department. 

Kind regards, 

on behalf of

Dr. Yih-Kuen Jan 

Academic Editor

PLOS ONE